# Virtual Receptors for Efficient Molecular Diffusion

**Matan Halfon, Eyal Rozenberg, Ehud Rivlin, Daniel Freedman**
Verily Research
Haifa, Israel
{matanhalfon,eyalrozenberg,ehud,danielfreedman}@google.com

## Abstract

Machine learning approaches to Structure-Based Drug Design (SBDD) have proven quite fertile over the last few years. In particular, diffusion-based approaches to SBDD have shown great promise. We present a technique which expands on this diffusion approach in two crucial ways. First, we address the size disparity between the drug molecule and the target/receptor, which makes learning more challenging and inference slower. We do so through the notion of a Virtual Receptor, which is a compressed version of the receptor; it is learned so as to preserve key aspects of the structural information of the original receptor, while respecting the relevant group equivariance. Second, we incorporate a protein language embedding used originally in the context of protein folding. We experimentally demonstrate the contributions of both the virtual receptors and the protein embeddings: in practice, they lead to both better performance, as well as significantly faster computations.

## 1 Introduction

**Background** In the realm of structural biology, a persistent challenge has revolved around the rational design of small molecules, particularly in the context of drug development. Specifically, the field of Structure-Based Drug Design (SBDD) is concerned with leveraging the three-dimensional structure of proteins to identify or craft new drug candidates capable of binding to the target protein, thereby inhibiting its activity. With the advent of precise protein folding models, such as AlphaFold [Jumper et al., 2021] and RosettaFold [Baek et al., 2021], accurate 3D protein structure has become widely available. This latter fact presents significant opportunities for advancing the field of SBDD, paving the way for the development of innovative algorithms designed to harness these 3D protein structures. Precise and efficient SBDD algorithms hold immense promise, promising to expedite and economize the process of drug development in the years ahead.

**Prior Work** Various ML approaches to SBDD have recently been developed. The pioneering work of Ragoza et al. [2022] employs a conditional Variational Autoencoder (VAE) on a discretized 3D atomic density grid. Following sampling, this grid structure undergoes a transformation into molecular structures through an atom fitting algorithm. A subsequent class of works including [Drotár et al., 2021] and [Peng et al., 2022] have adopted an autoregressive methodology. Here, the generation of molecular structures unfolds as a sequential process, with the probability distribution of each atom being learned in relation to the preceding atoms. Consequently, molecules are sampled step by step, with each step adding a new segment to the molecule, all within the confines of the binding site. Luo et al. [2021] introduce a model that captures the likelihood of a particular chemical element occupying a specific point in 3D space. Liu et al. [2022] propose the GraphBP framework, eliminating the need for spatial discretization. Instead, they generate atom types and relative locations while preserving equivariance. Other notable approaches include those based on recurrent neural networks [Zhang and Chen, 2022] and equivariant normalizing flows [Rozenberg and Freedman, 2023].

**SBDD via Diffusion** Diffusion models have become the leading generative approach in various domains, including images [Saharia et al., 2022], video [Ho et al., 2022], and audio [Kong et al., 2020].

NeurIPS 2023 AI for Science Workshop.

Recent advancements have witnessed the integration of diffusion models into the realm of structural biology. Example applications include the creation of novel molecules based on fragments [Levy and Rector-Brooks, 2023] and the de novo generation of entirely new proteins [Watson et al., 2022, Gruver et al., 2023]. Crucially to the current work, they have also been used in SBDD [Schneuing et al., 2022], producing some of the most promising results to date.

**Contributions** In this work, we expand on the diffusion approach to SBDD in two critical ways. **(1)** *Virtual Receptors*: A key issue in designing SBDD algorithms is the disparity in size between the drug/ligand and the target/receptor, as the former can often be several orders of magnitude smaller than the latter. There are two consequences of this disparity: learning can be challenging, as the model has a hard time focusing on the ligand; and the algorithm can be quite slow. To deal with these issues, we define the notion of a virtual receptor: a learned encoding model compresses the receptor graph to a much smaller graph, but which still retains key aspects of the structural information of the original receptor. We design the encoding model to respect the relevant group equivariance. **(2)** *Protein Language Embeddings*: A language model for protein sequences called ESM [Lin et al., 2023] utilizes billions of protein sequences to learn a powerful representation for these sequences. In the original work, this representation was used for predicting protein folding; here we use this representation as a useful embedding to provide high quality information on the amino acids. We experimentally demonstrate the contributions of both the virtual receptors and the protein embeddings: in practice, they lead to both better performance, as well as significantly faster computations.

## 2 Methods

### 2.1 Background: Diffusion Framework for Target-Aware Molecular Sampling

**Molecular Notation** A molecule has a given number of atoms, here denoted $n$. Each atom is specified by two items: (1) its coordinates in 3D, denoted by $x_i \in \mathbb{R}^3$; (2) its features, which will include the atom type, denoted by $h_i \in \mathbb{R}^d$. Note that in general the features will contain categorical and ordinal values. We will denote the coordinates and features of the entire molecule by $X = [x_1, \ldots, x_n] \in \mathbb{R}^{n \times 3}$ and $H = [h_1, \ldots, h_n] \in \mathbb{R}^{n \times d}$, respectively. These may be flattened and concatenated into a single vector describing the molecule, $z = \texttt{CONCAT}(\texttt{FLATTEN}(X), \texttt{FLATTEN}(H))$.

**Diffusion Models** A diffusion model involves two processes: a forward noising process, which takes the data and turns it into Gaussian noise; and a reverse denoising process, which takes Gaussian noise and turns it back into data. The forward process is specified by

$$q(z_t|z_{data}) = \mathcal{N}(z_t \mid \alpha_t z_{data}, (1 - \alpha_t^2)I) \tag{1}$$

where $t = 1, \ldots, T$, and $\alpha_t$ follows a set schedule from $\alpha_0 \approx 1$ to $\alpha_T \approx 0$ [Kingma et al., 2021]. Note that the process specified in Equation (1) is a variance preserving process [Song et al., 2020].

The reverse process is learned. It is known that

$$q(z_{t-1}|z_{data}, z_t) = \mathcal{N}(z_{t-1} \mid u_t z_t + v_t z_{data}, w_t I) \tag{2}$$

where $u_t, v_t, w_t$ are known functions of $\alpha_t$ and $\alpha_{t-1}$. This equation is interesting, but not useful practically as $z_{data}$ is unknown; thus, the diffusion model learns $z_{data}$ as a function of $z_t$. In practice, rather than learning $z_{data}$, one learns the noise from Equation (1); if $\hat{\varepsilon}_t$ is the estimated noise, then the data can be recovered as

$$\hat{z}_{data} = \frac{1}{\alpha_t} z_t - \frac{\sqrt{1 - \alpha_t^2}}{\alpha_t} \hat{\varepsilon}_t \tag{3}$$

The goal of learning is therefore to approximate the function $\varepsilon_\theta$ where $\hat{\varepsilon}_t = \varepsilon_\theta(z_t, t)$; training is achieved using an $L_2$ loss between the estimated noise and the ground truth noise. Inference can then be performed by repeatedly: (a) using $z_t$ to compute $\hat{\varepsilon}_t = \varepsilon_\theta(z_t, t)$; (b) computing $\hat{z}_{data}$ using Equation (3); and (c) sampling $z_{t-1}$ from Equation (2) using $\hat{z}_{data}$ in place of $z_{data}$.

**Equivariant Graph Neural Networks (EGNNs)** Our goal is now to learn the noise function $\varepsilon_\theta(z_t, t)$. In order to produce a final distribution which is invariant to both rigid transformations and permutations, we must design $\varepsilon_\theta$ to be equivariant to each of these transformations [Hoogeboom et al., 2022]. This can be achieved by using an Equivariant Graph Neural Network, or EGNN [Satorras

et al., 2021]. This is a message-passing GNN whose $s^{th}$ layer is specified as follows:

$$m_{ij}^s = \phi_e\left(h_i^s, h_j^s, \|x_i^s - x_j^s\|^2, t\right) \qquad\qquad m_i^s = \sum_{j=1}^N \phi_a(m_{ij}^s)m_{ij}^s$$

$$x_i^{s+1} = x_i^s + \left(\sum_{j \neq i} \frac{(x_i^s - x_j^s)}{\|x_i^s - x_j^s\| + 1}\right)\phi_x(m_{ij}^s) \qquad\qquad h_i^{s+1} = \phi_h(h_i^s, m_i^s) \qquad (4)$$

**SBDD via Conditional Diffusion Models**  Until now, we have described an unconditional generative model over molecules, i.e. $p(z)$. What we would like, however, is a conditional model of the form $p(z^\ell|z^r)$, where $z^\ell$ and $z^r$ are the vectors representing the ligand and receptor, respectively. This is easily achieved by replacing the noise estimator with a function which also depends on the receptor, i.e. $\hat{\varepsilon}_t = \varepsilon_\theta(z_t^\ell, z^r, t)$. This can be implemented via an EGNN over the combined graph of both the ligand and the receptor. To allow the network to distinguish between ligand and receptor atoms, one may append a length-2 one-hot vector to the initial feature vectors $h_i$.

## 2.2 Virtual Receptor Atoms

**Motivation**  As we have just seen, SBDD via diffusion can be achieved by learning the function $\varepsilon_\theta(z_t^\ell, z^r, t)$. A key issue in learning this function stems from the disparity in sizes between the ligand and the receptor: the latter is generally several orders of magnitude larger than the former. This leads to two problematic consequences. First, if the ligand and receptor are combined into a joint graph which is then used in the EGNN in Equation (4), the ligand can be "drowned out" by the much larger receptor, making learning challenging. (Note that this can be true even if the receptor's nodes are given by the $C_\alpha$ atoms instead of all of the atoms.) Second, inference and training are made considerably slower due to the presence of the large receptor. To deal with this issue, we introduce the notion of the *virtual receptor*: a smaller molecule which retains the properties of the receptor.

**Approach**  A given virtual receptor atom will be written as a convex sum of the true atoms

$$\tilde{x}_i^r(t) = \sum_{j=1}^{n^r} A_{ij}(X^r, H^r, t)x_j^r \quad i = 1, \ldots, \tilde{n}^r \qquad (5)$$

where a tilde will be used to denote virtual quantities, and $A_{ij}()$ is learnable. Thus, the virtual receptor atoms live within the convex hull of the true atoms. Note that we only generate $\tilde{n}^r$ such virtual atoms, where we take $\tilde{n}^r \ll n^r$. In particular, we suppose that $\tilde{n}^r = O(n^\ell)$, where $n^\ell$ is the number of ligand atoms. A similar equation holds for the features

$$\tilde{h}_i^r(t) = \sum_{j=1}^{n^r} A_{ij}(X^r, H^r, t)\xi_h(h_j^r) \qquad (6)$$

where $\xi_h$ can be an arbitrary function.

**Properties of $A_{ij}$**  To preserve the equivariance of the EGNN, we'll need for Equation (5) to be equivariant with respect to both rigid transformations and permutations. It is straightforward to see that E(3)-equivariance will hold if:

1. $A_{ij}(\cdot, H^r, t)$ is an $E(3)$-invariant function of the set of receptor atoms' positions $X^r$; and

2. $\sum_{j=1}^M A_{ij}(X^r, H^r, t) = 1$ for all $i$ (hence our requirement for the *convex* sum).

Permutation-equivariance will hold if $A_{ij}(\cdot, \cdot, t)$ is a permutation-invariant function of the set of receptor atoms' positions $X^r$ and features $H^r$.

**Implementation of $A_{ij}$**  The following architecture for $A_{ij}$ respects the above properties:

$$q_{jk}^r = \xi_q\left(\|x_j^r - x_k^r\|, h_j, h_k\right) \qquad b_j\,, \bar{b}_j = \texttt{SPLIT}\left(\xi_b\left(\frac{1}{n^r}\sum_{k=1}^{n^r} q_{jk}^r\right)\right) \qquad \bar{b}^{av} = \frac{1}{n^r}\sum_{j=1}^{n^r}\bar{b}_j$$

$$\hat{A}_{:j} = \xi_a\left(\texttt{CONCAT}(b_j, \bar{b}^{av}, \xi_k(h_j), \zeta(t))\right) \qquad\qquad A_{ij} = \frac{e^{\hat{A}_{ij}}}{\sum_{j'=1}^{n^r} e^{\hat{A}_{ij'}}} \qquad (7)$$

where $\hat{A}_{:j}$ is a vector of length $\tilde{n}^r$; the functions $\xi_q, \xi_b, \xi_a, \xi_k$, as well as $\xi_h$ from Equation (6), are all multilayer perceptrons; the function $\zeta(t)$ is an encoding of the time scalar, such as a sinusoidal encoding [Vaswani et al., 2017]; and SPLIT() splits the vector into two pieces.

**Initialization** In order to initialize the virtual receptor network in Equation (7), we use an autoencoding strategy. Specifically, we treat Equation (7) as an encoder; and the decoder is provided by switching the roles of the virtual and true atoms in Equation (7), i.e.

$$\breve{x}_i^r(t) = \sum_{j=1}^{\tilde{n}^r} \breve{A}_{ij}(\tilde{X}^r, \tilde{H}^r, t)\tilde{x}_j^r \quad i = 1, \dots, n^r \tag{8}$$

where the architecture for $\breve{A}_{ij}$ is the same as in Equation (7), but with different parameters. The loss between the original receptor atoms $x_i^r$ and their reconstructions $\breve{x}_i^r$ is a bipartite matching loss, i.e.

$$L(X^r, \breve{X}^r) = \min_{\pi \in \mathbb{S}^{n^r}} \sum_{i=1}^{n^r} \|x_i^r - \breve{x}_{\pi(i)}^r\|^2 \tag{9}$$

where $\pi$ is a permutation. The rationale for using this loss is that we should only expect the decoder to return the same *set* of atoms; however, the order within the set may be modified. We note that the optimization in Equation (9) may be solved with the Hungarian algorithm [Kuhn, 1955].

We make two further comments. First, we use $t = 0$ in both Equations (5) and (8) when training the autoencoder, as time is not relevant here (it is a diffusion-based variable). Second, we do not include reconstruction of the features $h$ as part of the initialization; we found that reconstruction of the positions gave a sufficiently good initialization.

### 2.3 Use of Protein Language Embeddings

The use of masked language model embeddings has had a profound impact across diverse domains. Specifically, the ESM (Evolutionary Scale Modeling) embedding [Lin et al., 2023] has emerged as a promising embedding for the protein domain, as it has demonstrated its capacity to encapsulate highly informative features for function prediction. In the original paper [Lin et al., 2023], it was used for protein structure prediction. Since then, it has found numerous other applications. Perhaps most similar in spirit the use in this paper is the use for de novo prediction of an entire complex of a protein and ligand (i.e. in this scenario the protein is not given but predicted) [Nakata et al., 2023]. Also worth mentioning is the use of the related AlphaFold network [Jumper et al., 2021] for SBDD [Chen et al., 2023]; however, note that in this paper only the protein sequence – not its structure – is given, and AlphaFold's use in this context is effectively about providing that structure.

In this paper, we leverage the ESM embedding for the purpose of injecting valuable insights about the overall protein structure. This augments our understanding of each amino acid's information in the context of its close neighborhood of amino acids and 3D structure. Specifically, we use the ESM embedding from the last layer of its encoder model as the new feature vector for the receptor node corresponding to a given amino acid (see Section 2.4). This adds key information about the protein residues' chemical and structural features that are affected by their proximal residues.

### 2.4 Implementation Details

Receptors are described by using only $C_\alpha$ atoms of each residue; in practice, we take a subset of the 100 closest $C_\alpha$ atoms to the center of mass of the binding site. We used 30 virtual receptor atoms, in order to be similar to the mean number of ligand atoms, which is 19.0 with a standard deviation of 6.8. In order to enable fast convergence for the autoencoder, we initialize the set of virtual receptor atoms with Farthest Point Sampling. Specifically, we begin with the atom closest to the center; and then for each subsequent iteration, we find the atom which is furthest from the entire set thus far constructed. After this initialization, the autoencoder is allowed to train in standard fashion. Details of the denoiser architecture are included in the Appendix.

## 3 Experiments

**Data** We use the CrossDocked dataset [Francoeur et al., 2020], which contains 22.5 million poses of ligands docked into multiple similar binding pockets across the Protein Database (PDB). In particular,

Table 1: Comparison of metrics for all methods. Best result is shown in **bold**, second best in underline. Note that inference time is measured in seconds per 100 molecules generated; and train time is measured in minutes per epoch (all methods were trained for 1,000 epochs).

| Model | QED (↑) | logP (↑) | Lipinski (↑) | SA (↑) | Novelty (↑) | Vina (↓) | Vina - top 10% (↓) | Inference Time (↓) | Train Time (↓) |
|---|---|---|---|---|---|---|---|---|---|
| Diffusion + VR + ESM | **0.465** | **2.34** | **4.84** | 0.78 | **0.98** | **-6.3** | **-7.2** | 48.7 | 4.4 |
| Diffusion + VR | 0.380 | 1.32 | 4.21 | **0.79** | 0.94 | -4.1 | -5.6 | 10.6 | **2.9** |
| Diffusion - Basline | 0.390 | 1.72 | 4.43 | 0.72 | 0.73 | -4.5 | -7.0 | 186.2 | 21.0 |
| GraphBP | 0.413 | 0.61 | 4.64 | 0.59 | 0.88 | -4.3 | -6.2 | **10.2** | N/A |
| CENF | 0.463 | 1.46 | 4.83 | 0.63 | 0.85 | -4.5 | -5.7 | 160 | N/A |

Table 2: Ablation study. ESM-X indicates the ESM model with X parameters. Best result is shown in **bold**, second best in underline. Inference time is measured in seconds per 100 molecules generated.

| Model | QED (↑) | logP (↑) | Lipinski (↑) | SA (↑) | Novelty (↑) | Vina (↓) | Vina - top 10% (↓) | Inference Time (↓) |
|---|---|---|---|---|---|---|---|---|
| Diffusion VR ESM-650M | **0.47** | **2.34** | 4.84 | 0.78 | **0.98** | **-6.3** | -7.2 | 48.7 |
| Diffusion VR ESM-150M | 0.43 | 1.82 | 4.54 | 0.71 | 0.94 | -5.6 | **-7.8** | 24.1 |
| Diffusion VR ESM-35M | 0.38 | 1.54 | **5.12** | 0.71 | 0.92 | -5.2 | -7.1 | 18.8 |
| Diffusion VR No ESM | 0.38 | 1.32 | 4.21 | **0.79** | 0.94 | -4.1 | -5.6 | **10.6** |

these docked receptor-ligand pairs all have binding pose RMSD lower than 2 angstroms. We use the authors' proposed split into training and validation sets. Based on considerations of training duration, we retain only those data points: (a) whose ligand has 30 atoms or fewer with atom types in {C, N, O, F}; (b) which have potentially valid ligand-protein binding properties, i.e. whose ligands do not contain duplicate vertices and whose predicted Vina scores [Trott and Olson, 2010] are within distribution. The refined datasets are of size 132,863 (train) and 63,929 (validation).

**Evaluation Methodology**   We compare our method to three competing techniques: a baseline diffusion method for SBDD [Schneuing et al., 2022]; GraphBP, an autoregressive method [Liu et al., 2022]; and a model based on conditional equivariant normalizing flows [Rozenberg and Freedman, 2023], which we denote CENF. The methods are compared using standard performance metrics, in particular: (i) QED (ii) logP (iii) Lipinski (iv) SA (synthetic accessibility) (v) Novelty (vi) Vina. These metrics are described in more depth in the Appendix. We also record compute metrics, in particular: (vii) inference time (viii) train time.

**Results**   The results are shown in Table 1. Note that in terms of the six performance metrics, our proposed method using both virtual receptors and ESM embeddings finishes top in five metrics; and second in SA, just barely behind our proposed method using only virtual receptors. In terms of inference time, the method with virtual receptors and without ESM embeddings is a factor of 17.6 times faster than ordinary diffusion, and is nearly as fast as GraphBP, an autoregressive technique. The addition of ESM embeddings requires more compute, yet the method is still 3.8 times faster than ordinary diffusion. A similar trend may be noted for train times.

**Ablation Studies**   We run our model under different settings, with the results shown in Table 2. In particular, we show the results of using different version of the ESM model for protein language embeddings; we show results without ESM, and with ESM models of 35M, 150M, and 650M parameters. In general, the trend is that performance improves as the size of the ESM model increases; while unsurprisingly, the inference time increases as well.

## 4   Conclusions

In this study, we have introduced the concept of virtual receptor atoms, a novel method for encoding a molecular graph structure into a more compact graph while retaining crucial structural information. By combining this technique with protein language embeddings based on ESM within a Structure-Based Drug Design diffusion model, we show experimentally that our performance surpasses both the baseline diffusion model and alternative methodologies rooted in autoregressive models and normalizing flows. This superior performance is evident across multiple standard metrics related to drug likeness and binding affinity; and also in terms of the speed of molecule generation. Our approach demonstrates promising potential in enhancing the capabilities of SBDD diffusion models for the generation of new drug candidates.

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

# A Appendix

## A.1 Architecture Details

A high level illustration of the scheme is shown in Figure 1.

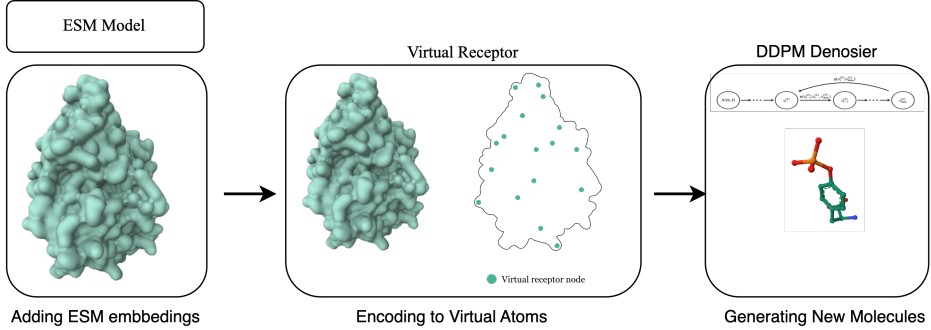

Figure 1: High level illustration of the proposed algorithm.

The denoiser model consists of 6 standard EGNN layers as shown in Equation (4) and described in [Satorras et al., 2021], where both the ESM embedding and the atom one hot vector are projected into the same dimension using a linear layer of dimension 256. The denoiser is trained with the AdamW optimizer [Loshchilov and Hutter, 2017] with a learning rate of $10^{-4}$ and a cosine decay scheduler for the denoising objective.

## A.2 Evaluation Metrics

We here describe each of the evaluation metrics used in Section 3. These are standard metrics for the field of SBDD.

(i) QED: a measure of drug-likeness with values $\in$ [0,1]; a higher QED indicates a greater likelihood for a molecule to be a successful drug candidate. QED is an aggregate measure which combines several desirable molecular properties [Bickerton et al., 2012].

(ii) logP: is defined as the octanol-water partition coefficient, which is a measure of lipophilicity (the ability of a molecule to mix with an oily phase rather than with water). logP values $\in$ [-0.4, 5.6] indicate molecules with increased potential to be viable drug candidates [Wildman and Crippen, 1999, Ghose et al., 1999].

(iii) Lipinski: is shorthand for Lipinski's Rule of Five [Lipinski et al., 1997, Veber et al., 2002], which is an empirically derived rule of drug-likeness.

(iv) SA: an acronym for synthetic accessibility [Ertl and Schuffenhauer, 2009]. The SA score, which takes on values in [0,1], measures how difficult is it to synthesize a given molecule, with higher values indicating that a molecule is more easily synthesized.

(v) Novelty: measures the fraction of entirely new molecules generated. A molecule is considered novel if its canonical SMILES string is not in the training set.

(vi) Vina: the Vina score [Trott and Olson, 2010] measures the binding affinity between a small molecule and their target pocket. It is an empirically derived measure for the binding affinity.

