# OpenReview forum: "Virtual Receptors for Efficient Molecular Diffusion"
_NeurIPS.cc/2023/Workshop/AI4Science — NeurIPS2023-AI4Science Poster_

### Official Review · Reviewer_Eg9v · 2023-10-25
**I think it is a decent paper and I am positive to its publication**

**Rating:** 6
**Confidence:** 2

**Review:**

The paper presents an innovative approach to Structure-Based Drug Design (SBDD) by integrating the concept of Virtual Receptors and Protein Language Embeddings. The blend of these two methodologies has the potential to advance the domain of SBDD, as the paper suggests. While I'm not an expert in this particular niche, I find the work could be useful based on its presented merits. I am positive about its acceptance, but I believe there are areas where the paper could benefit from additional clarity.

It would be beneficial to clearly specify the tasks or objectives associated with Tables 1 and 2. For instance, are these tables evaluating metrics related to the top 100 molecules as designed by the model? If so, it would be insightful to know which specific protein or proteins these molecules are intended to dock with. This added context would enhance the reader's understanding and the paper's overall coherence.

In Table 1, the term "VR" is used. It would enhance clarity if the full form, presumably "Virtual Receptor", is provided either in the table or its accompanying caption. This will ensure that readers unfamiliar with the abbreviation can easily follow the discussion and results.